# Welding Capabilities of Nanostructured Carbide-Free Bainite: Review of Welding Methods, Materials, Problems, and Perspectives

**Aleksandra Królicka [1,*], Andrzej Ambroziak [1] and Andrzej Żak [1]**

Faculty of Mechanical Engineering, Wroclaw University of Science and Technology, Wybrzeże Wyspiańskiego 27, 50-370 Wrocław, Poland

\* Correspondence: aleksandra.krolicka@pwr.edu.pl

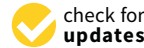

**Featured Application: Expanding the applications of nanostructured carbide-free bainitic steels by enabling them to be joined. This particularly applies to structures operating in difficult conditions (e.g., abrasive wear, contact fatigue, dynamic loads).**

**Abstract:** This article presents state-of-the-art welding methods and the weldability aspect of steels, particularly high-carbon nanobainitic (NB) steels, without carbide precipitates (CFB—carbide-free bainite). On the basis of research conducted to date, all welding methods with parameters and weld metals for NB CFB are presented. It was found that the process parameters significantly affected the mechanical properties of the welds, which were determined primarily by the properties of the low-temperature heat-affected zone. The microstructures of welded joints in the heat-affected and fusion zones are also described. The general requirements for welding processes, as well as problems and perspectives for further research, are presented.

**Keywords:** nanobainite; NB; CFB; high-carbon steels; high-strength steels; Si-rich steels; welding; HAZ; fusion zone

---

## 1. Introduction

Bhadeshia and Edmonds [1,2] were the first to introduce methods for designing steel that is characterized by a structure consisting of bainitic ferrite and austenite with high mechanical properties. A carbide-free structure is possible to obtain with a concentration of 2% Si [3,4]. In addition, the nanometric widths of the bainitic ferrite laths and film-like austenite (even with 20-40 nm, [5]) allow for an ultimate tensile strength (UTS) of 2.5 GPa to be reached with satisfactory ductility [6]. Austenite can be said to play an important role. It has a film-like morphology and is characterized by a high carbon concentration and high dislocation density and stability in comparison to blocky austenite [5,7]. Such materials are called nanostructured bainitic (NB) steels without carbide precipitates (carbide-free bainite (CFB)) and their development continues to this day (among others, recently published [8–10]). The high mechanical properties of nanostructured carbide-free bainite create the possibility of a significant reduction in mass, and also increase the durability of structural parts in many branches of industry. In addition to the high price, which is mainly associated with complex, long-term heat treatment and demanding chemical composition, the main limitation of the common use of NB and CFB in industry is their difficult welding [11]. A high carbon content in the range of 0.6/1.0 wt.%, silicon of 1.5/3.0 wt.%, and manganese of 0.6/2.0 wt.% allows for the refinement of the microstructure to nanometric values using a low temperature for the isothermal heat treatment [12], which is the main reason for the poor weldability of these steels. The fundamental problems that

reduce the mechanical properties of the welded joints are the formation of brittle martensite in the fusion zone, cementite precipitation in the heat-affected zone, and cold cracking [11].

## 2. Design of Weld Metals

Obtaining the structure of nanocrystalline bainite in an entire welded joint area determines the most favorable mechanical properties. The homogeneous bainitic structure is difficult to achieve during conventional welding methods. In addition, the NB CFB structure can be obtained for materials characterized by a specific chemical composition, which causes problems with the selection of weld metals. At present, there are no commercially available weld metals that allow for the structure of NB CFB in the weld zone. In Reference [13], the chemical composition of an electrode was proposed, which allows such a structure in the weld zone to be obtained with a small amount of martensite and retained austenite (Table 1). After the welding process, as a result of the high hardenability associated with the designed chemical composition of the weld material, the typical Widmanstätten structure was not achieved. High ultimate tensile strength (approximately 950 MPa) and yield strength (approximately 830 MPa) was not achieved in the joint. It was found that silicon does not significantly affect strength parameters but does have an influence on impact toughness. The increased concentration of silicon caused a decrease in the impact energy, which was explained by the slow development of martensite tempering in relation to the material with a lower content of silicon. Krishna Murthy et al. [14] proposed weld metals with a higher carbon content (0.32%), which, according to the authors, can replace the austenitic welds when welding armor steels. The used weld metals allowed the desired structure of CFB to be obtained with blocky austenite in the inter-dendritic areas. The presence of unstable blocky austenite reduces the impact toughness of the welds due to the induction of martensitic transformation under loads [15].

**Table 1.** Review of weld metals that allow a carbide-free bainite (CFB) structure to be obtained.

| Chemical Composition of Weld Metals wt.% | Process Parameters | Mechanical Properties | Microstructure and Comments | Reference(s) |
|---|---|---|---|---|
| (1) 0.10% C; 2.2% Mn; 0.86% Si; 2.1% Ni; 0.2% Mo (2) 0.12% C; 2.3% Mn; 1.38% Si; 2,1% Ni; 0,2% Mo (3) 0.1% C; 2.2% Mn; 1.63% Si; 2.1% Ni; 0.2% Mo | TIG Interpass temperature: 250 °C; Electrode ⌀4 mm; 30 runs; Current: 174 ± 1 A; Voltage: 25 ± 1 V; Input energy: 1.08 kJ/mm. | YS = 830 MPa UTS = 950 MPa The impact toughness decreases with increasing silicon content. No significant influence of silicon on weld strength was found. | Bainitic ferrite, austenite, and martensite in the weld. The structure is relatively homogeneous despite many weld beads. | [13] |
| 0.32% C; 1.6% Si; 1.6% Mn; 1.1% Ni; 1.1% Co; 1.0% Cr; 0.3% Mo | Shielded metal arc welding; Pre-heat: 350 °C; Current: 175 A; Voltage: 23 V; Input energy: 1.2 kJ/mm; Speed: 180 mm/min; Electrode ⌀4 mm; Electrode polarity: DCEN; Regeneration: 350 °C/6 h. | YS = 1010 MPa UTS = 1200 MPa Elongation: 14% Charpy Energy: 15 J | In the fusion zone FZ– the structure consists of bainitic ferrite and austenite (CFB). Blocky austenite in inter-dendritic areas. | [14] [16] |
| YS—Yield Strength | UTS—Ultimate Tensile Strength | DCEN—Direct Current Electrode Negative | | |

It was also found that the weld material with the structure of CBF was resistant to hydrogen cold cracking [14] and hot cracking [16]. The obtained ultimate tensile strength was 1200 MPa [14], but this is still insufficient in the context of high-carbon nanostructured bainitic steels (even at 2500 MPa [6]).

Therefore, weld materials with a higher strength should be designed. On the other hand, increasing the strength requires the carbon content to be increased, which causes worse weldability. Threat. Therefore, medium and high carbon bainitic steels were welded without weld metals in order to obtain high-strength parameters for the joints.

## 3. Review of Welding Methods

Over the last decade, attempts have been made to weld nanostructured CFB bainite. In Table 2, a review of welding processes of these steels that have so far been conducted is presented. The welding methods, chemical composition of the base materials, process parameters, mechanical properties, and a brief description of the microstructures are specified; in addition, comments are also provided. Furthermore, some welding methods involve steels with a high silicon content and a relatively low carbon concentration (0.34 wt.%). The table also includes some of the welding methods of steels after Quenching and Partitioning (Q&P) with regards to the chemical composition that favors the preparation of CBF structures.

Welding of high carbon bainitic steels (0.55/0.82%) was carried out, inter alia, using the TIG method [17–21]. It was found that the process parameters significantly influenced the microstructure and mechanical properties. A higher amount of delivered heat caused a higher volume of retained austenite in the weld (in which martensite was also found) [17]. The authors of Reference [17] applied preheating and also analyzed the welding process for steel in the softened delivery state. Due to the occurrence of cracks in welded sheets with a final bainitic structure, they proposed that it should be welded in the delivery state and that the heat treatment process should be carried out afterwards. Fang et al. [19–21] carried out a welding process using the regeneration technique. Regeneration involves controlled cooling of the welded joint to the isothermal heat treatment temperature and annealing at this temperature for a sufficient time to allow completion of bainite transformation. After welding with regeneration, a bainitic structure with coarse retained austenite was obtained [19]. No cold cracks or martensitic structures were found in the weld zone. After a tensile strength test, fractures were located in the heat-affected zone, where the presence of cementite was identified [19]. In order to accelerate the beginning of the bainite transformation, use of deformation in the welded joint was suggested by applying the rotary impacting trailed welding (RITW) method [22]. In this method, the welding and impacting occurs synchronously, which makes it possible to introduce compressive and shear stresses in the material at the same time during the welding process. Bainitic steel after heat treatment was also welded using the TIG method. The rotary impacting head was placed 30 mm behind the welding gun and 5 mm away from the welding centerline, which, in this area, corresponds to a temperature of 600°C. However, in this method, heat treatment (regeneration) is necessary after the welding process. It was found that the RITW method allows for a reduction in the time necessary for bainite transformation in the deformed austenite zone, but the used regeneration time, however, was not long enough to complete the transformation. In addition, the deformation of austenite before phase transformation affected the morphology of bainite—the bainitic ferrite laths were arranged in the direction of deformation. In Reference [23], two-pass welding was used to refine the austenite grains. The authors proposed such a process due to the fact that fine austenite grains will accelerate the nanobainitic transformation [24,25], such as in the case of the material used in this study [23]. Acceleration of bainitic transformation will shorten the regeneration time, which will reduce the cost of heat treatment and increase the process' technological efficiency. As with the previous method, the impacting trailed welding (ITW) method uses a first impacting head placed 30 mm behind the welding gun, and the second welding gun is 60 mm behind the first welding gun in the welding direction (Figure 1). Regeneration of the weld joint involved two stages, and the cycles are shown in Figure 2. Different recrystallization temperatures were used, ranging from 700 °C, 750 °C, and 800 °C

with times of 10 s, 30 s, 50 s, and 100 s. It was found that the conducted static recrystallization process in the impact zone reduced the grain size from 106 ± 42 μm to 36 ± 13 μm. The recrystallized grains were also smaller than the base material, where the grain size was around 50 μm. In addition, it was found that a higher volume of bainitic ferrite was obtained in the recrystallized grains, which also confirmed the reduction of the bainite transformation time and the same time of regeneration. Welded joints after the ITW process showed a higher ultimate tensile strength (2100 MPa) when compared to the process without ITW (1400 MPa) [23].

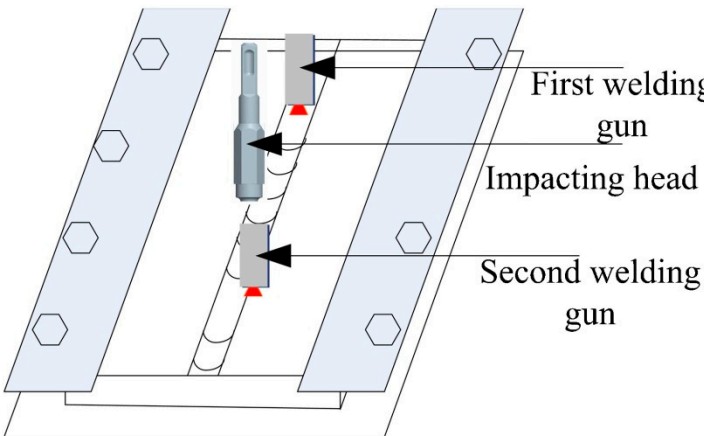

**Figure 1.** Scheme of the welding process using the two-pass impacting Trailed welding (ITW) method [23].

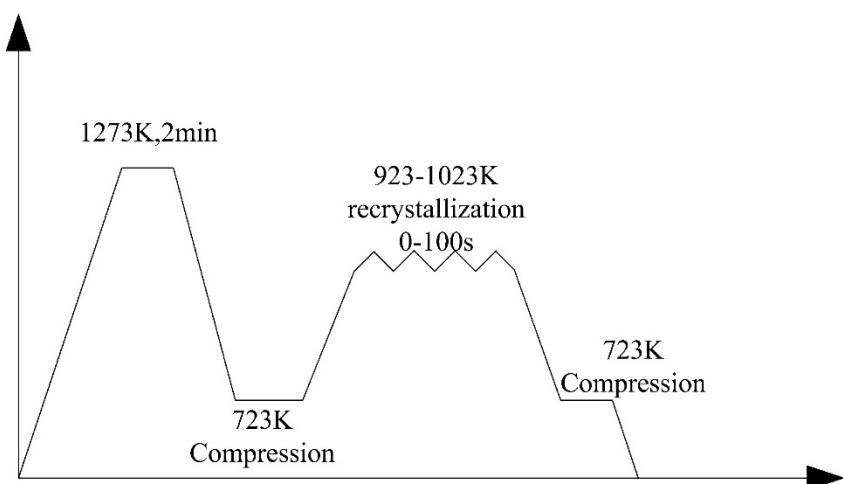

**Figure 2.** Two-pass regeneration with static recrystallization [23].

In Reference [26], for welding high carbon bainitic steels, laser welding with post-weld rapid heat treatment (PWRHT) is proposed. When the welded joint after the welding process approaches the beginning of the martensitic transformation (Ms), it is quickly heated to a temperature lower than $A_1$ (in less than 10 seconds). As a consequence, disadvantageous, brittle martensite in the welds and cold cracks are avoided. The structure after laser welding and PWRTH consisted of ferrite, austenite, and cementite, and its hardness was comparable to the base material. The obtained structure was different to the correct bainitic structure and, despite a comparable hardness level, it showed lower mechanical properties. Fang et al. [21] proposed a laser beam welding process with regeneration, similar to the TIG method. It was found that low input energy (60 kJ/m) in laser welding, when compared to the TIG method where high input energy was used (908 kJ/m), allows for the achievement of high ultimate tensile strength (33 MPa lower than the strength of the base material). The high tensile strength of

welded joints is due to the high stability of the microstructure in the heat-affected zone at low input energy processes.

Another welding method used for nanostructured CFB steels is friction stir welding (FSW) [27,28]. The joining process takes place in a solid state, which reduces the formation of distortions and defects associated with thermal contraction. In the FSW method, a rotating tool containing the shoulder and pin is plunged into the joint between the two front-mounted plates. As a result of friction and plastic deformation of the pin on the surface of the plates, heat is generated, which causes the softening of the material in the area of the tool. The softened material then flows to the rear edge of the shoulder, where it is stirred. The material after cooling forms a joint between the plates [29]. In References [27,28], it was found that the parameters of the FSW process affect the mechanical properties and microstructure of the obtained joints. The research concerned silicon-rich steels with a relatively low carbon content (0.34 wt.%). The increase in rotational speed increases the hardness of the zone, defined in the work as the stir-zone (SZ), located in the area of the moving shoulder. However, the hardness obtained at the lowest rotational speed was also high when compared to the thermo-mechanical affected zone (TMAZ) and base material. This increase in hardness is explained by the appearance of martensite under the influence of the thermo-mechanical effect (TRIP) and suggests the use of heat treatment after the joining process. However, the presence of martensite was not confirmed and the change in the volume of retained austenite in this zone was not determined. Scanning electron microscopy does not allow microstructures to be unequivocally identified. Carbide precipitates, which are likely in the TMAZ zone, were not analyzed.

In Reference [18], an attempt was made to friction weld NB CFB steel with C45 steel. The parameters of the welding process are presented in Table 2. No cracking was found in the Fusion zone, while microcracks were identified in the heat-affected zone (HAZ). When joining C45 steel with bainitic steel in the delivery state (perlite and ferrite), no microcracks were found [18]. However, welding steel in the delivery state requires heat treatment after the process in order to obtain a bainitic structure, which is an additional technological process (it will also change the mechanical properties of the C45 steel).

Wang and Speer [30] described the welding capabilities of commercial QP980 steel after quenching and partitioning (Q&P) heat treatment. The Q&P process was proposed by Speer et al. [31] and allows a high-strength martensite–ferrite–austenite structure for C–Si–Mn/C–Si–Mn–Al steels to be obtained. Heat treatment consists of complete or partial austenitization and cooling to the temperature between the beginning and the end of the martensitic transformation ($M_s$ and $M_f$), followed by isothermal anealing in the higher temperature range of martensitic or bainitic transformation [32,33]. Due to the high concentrations of silicon and aluminum in these steels, precipitation of cementite is inhibited, and the structures are therefore carbide free. Carbon enriches (above all) austenite and ferrite [33]. Heat treatment parameters determine the proportions of phases in the structure in which CFB bainite occur [33]. For this reason, this review also included welding methods for steel after Q&P treatment. The microstructure of the tested QP980 steel contained martensite, austenite, and ferrite, and no carbide-free bainite was identified [30]. However, the chemical composition of QP980 steel favors the structure of carbide-free bainite Therefore, the presented welding methods can be useful for planning the welding process of bainitic steels with a similar chemical composition. Steel after Q&P treatment was welded using resistance spot welding (RSW), laser welding, and the MAG method (Table 1). The authors assessed the weldability of QP980 steel as good and all welding methods were successfully carried out (i.e., no excessive welding defects and good mechanical properties) [29]. Resistance spot welding (with thermal simulation) was also performed for steel with a higher carbon concentration (42SiCr) after Q&P treatment. The satisfactory mechanical properties (shear strength) of the welded joints were not obtained and, in order to reconstruct the mechanical properties of the base material, the authors proposed the future use of a regenerative treatment in conditions similar to the Q&P parameters [34]. This result confirms the difficulty of welding steel with a higher carbon concentration.

**Table 2.** Review of welding methods for nanostructured carbide-free bainite and for selected steels after quenching and partitioning(Q&P) heat treatment.

| Welding Methods | Chemical Composition of Base Material wt.% | Process Parameters | Mechanical Parameters | Microstructure and Comments | Reference(s) |
|---|---|---|---|---|---|
| TIG | 0.82% C; 1.2% Si; 2.5% Mn; 0.8% Mo; 1.8% Cr; 1.5% Al; 1.0% Ni | 1. Current: 160 A; Voltage: 16 V; Speed: 22 mm/min; Sheet thickness: 6.0 mm 2. Current: 160 A; Voltage: 15 V; Speed: 50 mm/min; Sheet thickness: 6.0 mm 3. Current: 150 A; Voltage: 15 V; Speed: 80 mm/min; Sheet thickness: = 4.0 mm | UTS up to 60% of the base material; 1. UTS = 850 MPa 2. UTS = 1200 MPa 3. UTS = 1578 MPa | A higher amount of heat caused a higher volume of retained austenite. A small volume of martensite in the weld and HAZ. No preheating and regeneration. | [17] |
| TIG | (1) 0.61% C; 1.5% Mn; 1.7% Si; 1.3% Cr. (2) 0.55% C; 1.9% Mn; 1.8%Si; 1.3% Cr; 0.8% Mo. | Pre-heat = 230 °C; Current: 100 A (impulse 160 A); Impulses frequency: 30 Hz; Speed: 100 mm/min; Sheets thickness: 5.0 and 8.0 mm Cover gas: 100%Ar; Gas flow: 15 L/min. | Base material: Hardness: 640 HV. Softened state + heat treatment: 590/610 HV. Maximum value of nanobainitic welds: 720 HV in the HAZ | The sheets welded in the softened state did not show cold racks. Cold cracks were identified in welded joints of steel with a nanobainitic structure. | [18] |
| TIG + regeneration | 0.76% C; 1.0% Si; 1.3% Cr; 1.0% Mn; | Current: 140 A; Voltage: 20 V; Speed: 185 mm/min; Samples: 2 × 40 × 100 mm Regeneration: 250 °C /5 days | Base material: UTS = 1950 MPa, A = 2% After welding: UTS = 1410 MPa, A = 0.8% | No cold cracks in the weld. After regeneration, the weld had a bainitic structure and a small amount of retained austenite. Cementite precipitates were identified in the HAZ. | [19] [20] |
| TIG + regeneration | 0.82% C; 1.7% Si; 2.0% Mn; 0.2% Cr; 0.4% Mo; 1.1% Ni; | Input energy: 908 kJ/m Speed: 18.5 mm/min Sheets thickness: 2 mm Regeneration: 250 °C/5 days | Base material: UTS = 1877 MPa After welding: UTS = 1680 MPa | Precipitates of cementite in HAZ, which increased the width of bainitic ferrite laths, austenite decomposition in the LHAZ. | [21] |

**Table 2.** *Cont.*

| Welding Methods | Chemical Composition of Base Material wt.% | Process Parameters | Mechanical Parameters | Microstructure and Comments | Reference(s) |
|---|---|---|---|---|---|
| RITW (TIG + rotary impacting head) + regeneration | 0.87% C; 1.2% Si; 1.5% Mn; 0.3% Mo; 0.5% Ni; 1.1% Al | Current: 140 A; Voltage: 18 V; Speed: 90 mm/min Samples: 10 × 80 × 100 mm Regeneration: 1. 250 °C/1.5 h 2. 250 °C/2.5 h | | The RITW process accelerated the bainite transformation. Bainitic ferrite in the deformed austenite zone was arranged in accordance with the direction of deformation. | [22] |
| ITW (TIG + two-pass impacting head) + regeneration | 0.87% C; 1.2% Si; 1.5% Mn; 0.3% Mo; 0.5% Ni; 1.1% Al | Current: 220 A; Voltage: 18 V; Speed: 2 mm/s Samples: 10 × 80 × 100 mm Argon flow: 2 L/min Recrystallization: 700 °C; 750 °C; 800 °C 10 s; 30 s; 50 s; 100 s. Regeneration: 250 °C/2 h | Without ITW: UTS = 1400 MPa Elongation: 2% With ITW: UTS = 2010 MPa Elongation: 3/4% | Grain size: Base material: ~50 μm Without ITW: 106 ± 42 μm With ITW, recrystallization area: 36 ± 13 μm. The ferrite volume was higher for recrystallized grains and the bainite transformation time was reduced. | [23] |
| Laser Welding + PWRHT | 0.78% C; 1.0% Si; 1.5% Mn | PWRHT: Cooling of the welded joint (T > Ms); Heating with speed 10 s (T < Ac1); cooling to RT. | | No cold crack in the weld. HAZ: ferrite, cementite, and austenite. | [26] |
| Laser Beam Welding + regeneration | 0.82% C; 1.7% Si; 2.0% Mn; 0.2% Cr; 0.4% Mo; 1.1% Ni; | Input energy: 60 kJ/m Speed: 100 mm/min Sheet thickness: 2 mm Regeneration: 250 °C/5 days | Base material: UTS = 1877 MPa After welding: UTS = 1844 MPa | Slight changes in LHAZ. A small amount of cementite, austenite partly decomposed. | [21] |
| Friction Stir Welding | 0.34% C; 1.8% Mn; 1.5% Si; 0.9% Cr; | Stir speed: 80; 100, 150, 200 rpm Feed rate: 35 mm/min | Hardness of base material: 425 HV Hardness of stir-zone: 650–725 HV | In the stir zone, there was an increase of hardness in relation to the thermo-mechanical affected zone. Thermoplastic deformations caused the transformation of austenite into martensite. | [27,28] |

**Table 2.** *Cont.*

| Welding Methods | Chemical Composition of Base Material wt.% | Process Parameters | Mechanical Parameters | Microstructure and Comments | Reference(s) |
|---|---|---|---|---|---|
| Quenching and Partitioning + Resistance Spot Welding | QP980 steel 0.15–0.30% C; 1.5–3.0% Mn; 1.0–2.0%Si; Electrode: ISO 5821:2009 type B; ⌀4 mm | Weld force: 5.8 kN, 1305 lbf Weld pulse: 3 Sheet thickness: 1.6 mm Cooling: 2 L/min; 0.5 gal/min | Hardness of base material: 300 HV. After welding: 473–512 HV. Shear strength increases with current. | Welding defects such as cracks, pores or shrinkage cavities were not identified. The authors assessed the overall shear strength of welded joints as good. | [30] |
| Quenching and Partitioning + Laser Welding | QP980 steel 0.15–0.30% C; 1.5–3.0% Mn; 1.0–2.0% Si; | Energy: 3 kW Speed: 5 m/mm Cover gas: He Gas flow: 15 L/min Sheet thickness: 1.6 mm | After welding: UTS = 1081 MPa (Fracture in area of base material) Elongation: 7.3% (base material: 10.3%) | No welding defects. Weld area: low-carbon martensite. HAZ: martensite and ferrite. | [30] |
| Quenching and Partitioning + MAG | QP980 steel 0.15–0.30% C; 1.5–3.0% Mn; 1.0–2.0% Si; Electrode: ER110S | Energy: 3.6 kJ/cm Speed: 35 cm/min Cover gas: 80%Ar + 20% $CO_2$ Gas flow: 14 L/min Sheet thickness: 1.6 mm | After welding: UTS = 991 MPa Hardness: <500 HV | No visible welding defects or softened areas. | [30] |
| Quenching and Partitioning + Resistance Spot Welding+ simulation | 0.42% C; 2.0% Si; 1.3% Cr; 0.8% Mn; | Welding Energy: 3.13/3.39 J Nugget diameter: 7.26/7.33 mm Sheet thickness: 1.2 mm | Base material: UTS = 1841 MPa YS = 1030 MPa Hardened zone: 700 HV. Softened zone in HAZ: 370 HV. After welding: Shear strength: 7–10 kN | The maximum process temperature had a stronger effect on hardness than heating speed and heating time. | [34] |
| UTS—Ultimate Tensile Strength | YS—Yield Strength | HAZ—Heat-Affected Zone | PWRHT—Post-Weld Rapid Heat Treatment | RITW—Rotary Impacting Trailed Welding ITW—Impacting Trailed Welding | |

## 4. Analysis of Microstructure Changes after the Welding Process

The mechanical properties of welded joints are related to the microstructure of welds and the heat-affected zone. During welding processes, both the microstructure and grain size of the prior austenite change as a function of the distance from the weld face. A thorough analysis of structural changes may enable optimization of the welding parameters, which in turn will minimize zones with adverse mechanical properties.

The following zones were distinguished in welded joints (Figure 3):

- Low-temperature heat-affected zone (LHAZ): between $A_1$ temperature and the base material;
- High-temperature heat-affected zone (HHAZ): between the fusion line and $A_1$ temperature;
- Fusion zone (FZ): located at the weld area.

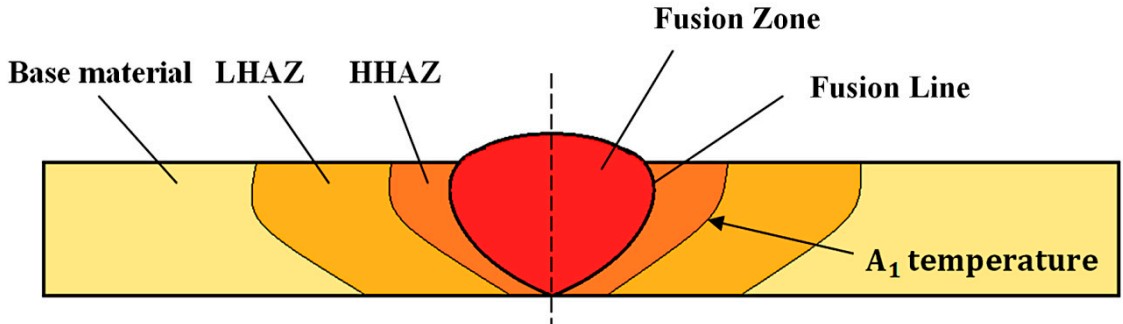

**Figure 3.** Scheme of welded joint.

### 4.1. Low-Temperature Heat-Affected Zone

One of the main problems of welding nanostructured bainitic steels is the precipitation of cementite in low-temperature heat-affected zones (LHAZs). Analogous to tempering processes, there are areas located in the LHAZ that correspond to tempering at different temperature ranges. In Reference [6], it was found that tempering at temperatures of up to 500 °C did not cause significant changes in the hardness and microstructure of nanobainitic steels. However, during longer period of heating, precipitation of cementite may occur at temperatures above 450 °C [35–38]. Areas of blocky austenite at higher temperatures (above 550 °C) were decomposed to fine dispersive perlite, and austenite with a film-like morphology was decomposed to ferrite and cementite due to the limited volume for pearlite growth [38]. In this temperature range, the austenite completely decomposed [38]. Further increases of the tempering temperature resulted in a ratio of ferrite and cementite that corresponded to the equilibrium state and an increase in the width of the ferrite plates [39], which reduced the mechanical properties. However, according to Podder and Bhadeshia [37], long-term tempering also causes a significant decrease in the volume of austenite, which becomes less stable by the precipitate of cementite. Then, austenite with a lower concentration of carbon was transformed to martensite after cooling to an ambient temperature. There was also no segregation of alloying elements between the ferrite/austenite phases during tempering [36]. In addition to research on the nanostructured CFB bainite stability during tempering, in situ tests were carried out during the continuous heating of two silicon-rich steels [40]. It was found that for a steel of 0.84C–2.26Mn–1.78Si–0.25Mo–1.55Co–1.47Cr, austenite began to decompose at 400 °C, while for the alloy 1.04C–1.97Mn–3.89Si–0.24Mo–1.43Al, decomposition began to occur at 580 °C. Therefore, the higher contents of silicon and aluminum in the steel caused a higher stability in the austenite. In addition, tests performed during continuous heating were closer to welding conditions than tempering over longer time periods. The differences in the obtained results indicate the need for further research, especially during continuous heating. In the context of welding processes, cooling to an ambient temperature or a regeneration temperature

after subsequent heating stages is also important due to the fact that the mechanisms of austenite decomposition can be significantly different.

Fang et al. [21] conducted research using a thermomechanical simulation of the LHAZ zone at 400 °C for low- and high-energy welding with regeneration (Figure 4). The samples after reaching 400 °C were cooled to the regeneration temperature (250 °C). In the initial steel consisting of 0.82C–1.66Si–2.05Mn–0.22Cr–0.36Mo–1.06Ni, a typical nanobainitic structure consisting of thin ferrite laths and austenite with a film-like morphology (and a small fraction of blocky austenite) was also identified (Figure 5a). For the low-energy process, no significant differences were found when compared to the base material. It was found that only short austenite structures with a film-like morphology were decomposed to spheroidal precipitations of austenite (Figure 5b). In the case of high-energy welding processes, the microstructure changed substantially. The short film-like austenite completely decomposed, while the areas of long filmy austenite became discontinuous. Many spheroidal precipitations also formed in the bainitic ferrite matrix (Figure 5c). According to the authors, the blocky austenite in a high-energy welding process decomposes into a fine nanostructured bainite with a different crystallographic orientation than the nearby bainite (Figure 5d). On the basis of investigations using transmission electron microscopy, no precipitates of cementite were found in the nanostructured bainite in the base material (Figure 6a). For the low-energy process, a small amount of cementite located at the ferrite–austenite interface was identified (Figure 6b). For the high-energy process, a large amount of cementite was found in the bainitic ferrite matrix with clearly increased widths of the laths (Figure 6c) [21]. However, in the presented research, the identification of cementite and its crystallographic orientation to the matrix using electron diffraction was not determined. This would have provided knowledge on the precipitation mechanism in steels with high carbon concentrations.

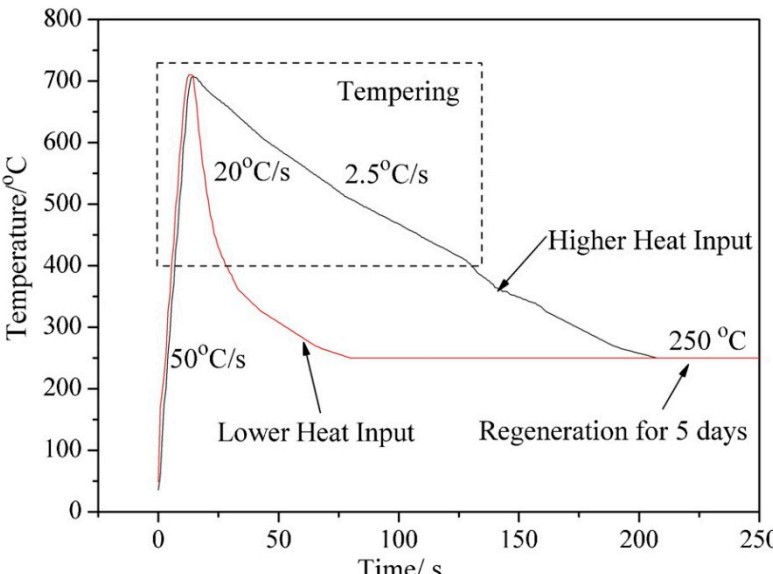

**Figure 4.** Thermal cycles simulation of the low-temperature heat-affected zone for low- and high-energy welding processes [21].

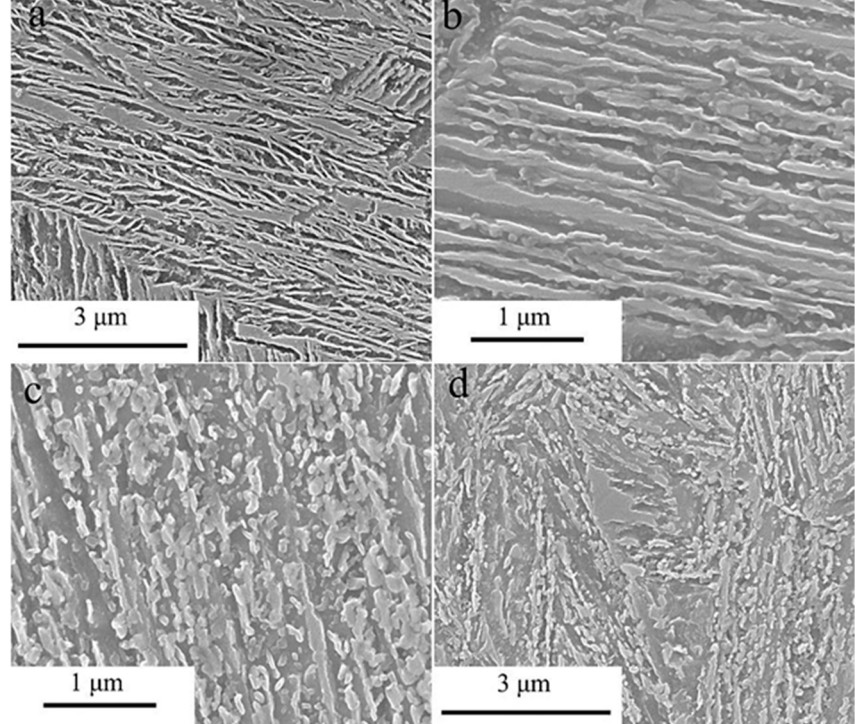

**Figure 5.** Microstructure of 0.82C–1.66Si–2.05Mn–0.22Cr–0.36Mo–1.06Ni steel. (**a**) Initial microstructure; (**b**) film-like morphology of the microstructure of the simulated low-energy welding process; (**c**) degraded microstructure for the simulated high-energy welding process; (**d**) area of blocky austenite for the high-energy welding process. Images produced via SEM [21].

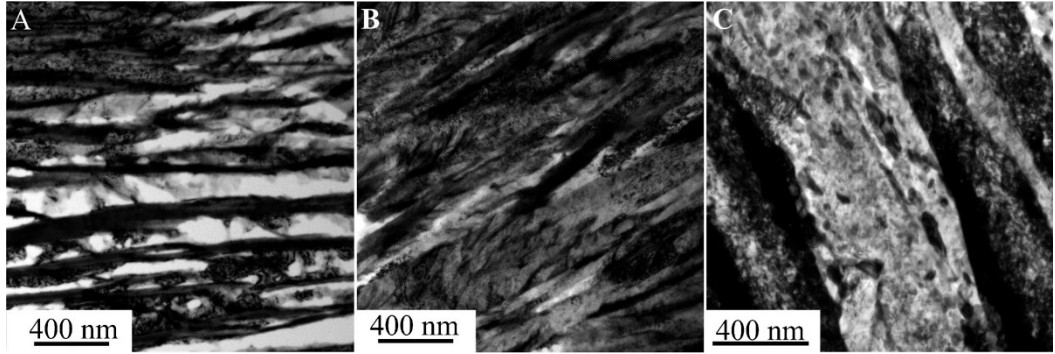

**Figure 6.** Bright field image of 0.82C–1.66Si–2.05Mn–0.22Cr–0.36Mo–1.06Ni steel. (**a**) Base material; (**b**) low-energy welding process; (**c**) high-energy welding process. Images produced via TEM [21].

In References [19,21], it was found that the fracture after tensile strength tests was located in the LHAZ, which indicates a significant impact of structural changes on this area. Welding and regeneration parameters significantly affect the phase morphology after the welding process. Fang et al. [20] stated that low-input energy processes are more advantageous when compared to high-input energy processes, which is confirmed, apart from significant changes in the microstructure, by the results of tensile strength tests of welds. For this reason, further research is necessary in order to reduce the most unfavorable zones and to develop analyses of the mechanisms of decomposition in nanocrystalline CFB structures in real welding process conditions.

## 4.2. High-Temperature Heat-Affected Zone and Fusion Zone

In the high-temperature heat-affected zone and fusion zone, as a result of high temperatures (above $A_1$), recrystallized austenite is formed, which during the rapid cooling after the welding process

is transformed into brittle martensite. This is the main reason for the cold cracking of these steels [19]. Thus, the following analysis includes welds after regeneration aimed at reconstructing the structure of the base material.

In the fusion zone, the columnar solidified structure characteristics for welding processes were identified. The visible structure in the inter-dendritic areas differs from the structure of the base material due to the segregation of carbon, chromium, and manganese [20]. In References [14,20], it was found that unstable blocky austenite occurs in the inter-dendritic areas. It was also determined that in the FZ zone, in addition to the coarse inter-dendritic structure, there is also a typical fine nanobainite structure and blocky austenite [20]. However, the volume of austenite in the weld is lower than in the base material [20], which can be explained by the reduction of the carbon content during welding [41] and, thus, by the increased rate of bainite transformation.

The presence of less stable, coarse blocky austenite in both the FZ and HHAZ is the reason for a significant reduction of impact toughness in relation to the base material [15,42,43]. On the other hand, blocky austenite increases elongation due to the TRIP effect [6]. A more preferred morphology is stable film-like austenite because it increases strength and fracture toughness. For this reason, a large amount of blocky retained austenite in the FZ and HHAZ is undesirable. The temperature of regeneration can affect the amount of retained austenite. It is known that increasing the temperature of bainitic transformation increases the amount of retained austenite [44,45], in which case the welded joints may increase the volume of unstable blocky austenite [20].

## 5. Conclusions

On the basis of current scientific knowledge, the methods of welding nanostructured bainitic steels commonly used today were determined. This study focused on obtaining high strength parameters of welded joints, often comparable with the base material. Problems and prospects for further research were also presented.

### 5.1. Welding Process

The conducted research on the possibility of welding high-carbon nanostructured CFB steels allowed for the determination of the requirements for welding parameters in order to achieve the high-strength parameters of welded joints without weld metals:

- Before welding processes, pre-heating should be used at a temperature close to that of the bainitic transformation. This prevents cold cracking and allows for control of the cooling process after the welding process is finished.
- Welding process parameters should be characterized by low-input energy, which results in more advantageous structural changes in the HAZ, less blocky austenite in the weld (fusion zone and high-temperature affected-zone), and higher strength compared to high-input energy processes.
- In order to obtain high mechanical parameters, comparable to the base material, heat treatment should be performed after the welding process which will reconstruct the NB CFB structure in the welded joint. Regeneration should be designed similar to the previously conducted heat treatment on the base material. It should be noted that an appropriate regeneration design should be based on dilatometric research, because the regeneration technique requires the determination of real phases of transformation temperatures and times.
- Long-term regeneration can be reduced by introducing deformation into the FZ, HHAZ, and grain refinement. However, the implementation of these methods requires further research and the design of appropriate equipment.
- The highest mechanical properties of welded joints can be obtained after welding in the delivery (softened) state, and then after the conducted isothermal heat treatment that is aimed at obtaining a nanocrystalline structure. There are no problems with the precipitation of cementite in the heat-affected zone due to the complete recrystallization of the weld and the base material. However,

this requires an additional technological process. In addition, due to the fact of their dimensions, not all welded constructions can be heat treated.

### 5.2. Problems and Perspectives

Welding of high-strength and high-carbon nanobainitic steels is difficult in terms of many factors, including welding parameters, heat treatment, choice of weld-metals, and microstructure. The main problems, as well as prospects resulting from the conducted research, were identified:

- The weld materials so far proposed have shown satisfactory resistance for cold cracking when using pre-heating, an ultimate tensile strength of up to 1200 MPa, and a structure consisting only of austenite and ferrite phases in the weld. However, welding of high-carbon steels requires stronger weld metals. Designing stronger weld metals, including those with a higher carbon concentration, would allow for easier joining and a wider range of welding methods for nanobainitic steels, which, until now, were welded without weld materials.

- The weakest zone of welded joints is the low-temperature affected-zone (LHAZ), where softening occurs due to the decomposition of austenite. Welding parameters directly affect the resumption of bainitic structures to the equilibrium state. However, the mechanisms of the austenite decomposition process and the precipitation of cementite require further research. This research should also involve real welding conditions. The development of a theory regarding these processes could allow for the reduction of unfavorable zones.

- A disadvantage which is difficult to avoid is the grain growth due to the influence of high temperatures, which results in lower mechanical properties. Analysis of the grain growth of bainitic steels (containing silicon and the lack of elements forming carbides inhibiting the growth of austenite grains) has not yet been described in detail.

- The required regeneration time to complete the bainite transformation is long (up to a few days), which is a problem in industrial applications. Shortening the time requires further research regarding the design of the chemical composition of base materials and weld metals, and also improvement of the deformation process (austempering) and the kinetics of phase transformations that depend on the regeneration parameters and grain size.

- To date, no residual stress research [46] on NB CFB welded joints has been presented. Due to the complex welding process and the focus on maximizing mechanical properties, the effect of residual stressors can be significant.

- Investigations on the mechanical properties of NB CFB welded joints did not include the aspect of fatigue. Due to the possibility of using welded structures during fatigue conditions, such research should be carried out in the future.

**Author Contributions:** A.K.; writing—original draft preparation, A.Ż.; writing—review and editing, A.A.; supervision.

**Funding:** POWR.03.02.00-00-I003/16.

**Acknowledgments:** Figures 1 and 2 copyright © Institute of Materials, Minerals and Mining reprinted with permission of Taylor & Francis Ltd. (http://www.tandfonline.com) on behalf of the Institute of Materials, Minerals and Mining. Song, K.J.; Fang, K.; Yang, J.G.; Ma, R.; Liu, X.S.; Wang, J.J.; Fang, H.Y., "Acceleration of regeneration treatment for nanostructured bainitic steel welding by static recrystallization", Materials Science and Technology. Figures 3–5 copyright © reprinted with permission of the publisher Taylor & Francis Ltd. Fang, K.; Yang, J.G.; Song, K.J.; Liu, X.S.; Wang, J.J.; Fang, H.Y., "Study on tempered zone in nanostructured bainitic steel welded joints with regeneration", Science and Technology of Welding and Joining, 26 June 2014.

**Conflicts of Interest:** The authors declare no conflict of interest.

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
