# Peer review of "Welding Capabilities of Nanostructured Carbide-Free Bainite: Review of Welding Methods, Materials, Problems, and Perspectives"

_applsci, doi:10.3390/app9183798_

Round 1

Reviewer 1 Report

According to the title, this manuscript is expected to be focused on the welding of nanostructured carbide-free bainte steels. While the authors explained proper reasons, adding contents related to the Q&P steel welding is somewhat confusing.

Table 2 includes some results of friction stir welding. While this content is explained briefly in section 3 of "Review of Welding Methods," in Section 4 of "Analysis of microstructure changes after the welding process," no discussion about Stir-Zone and thermo-mechanical affected zone is provided.

Except for Table 2, the difference of mechanical properties between base metal and weldment is discussed.  

Author Response

Dear Reviewer,

Thank you for your review of our manuscript and few important comments. We hope that with their help we were able to improve the quality of the article. We have compiled your comments and our responses below:

1. "According to the title, this manuscript is expected to be focused on the welding of nanostructured carbide-free bainte steels. While the authors explained proper reasons, adding contents related to the Q&P steel welding is somewhat confusing."

Response 1: Of course, the article concerns nanostructured carbide-free bainitic steels. Steels after Q&P treatment were added due to their chemical composition, derived from carbide-free bainitic steels. After suitable Q&P treatment, carbide-free bainitic structures are also possible to obtain. For this reason, we believe that such information may be useful for readers in the context of the analyzed topic. We highlighted that this is additional information that does not apply directly to nanobainitic steels: “The microstructure of the tested QP980 steel contained martensite, austenite and ferrite, and no carbide-free bainite was identified [27]. However, the chemical composition of QP980 steel favors the structure of carbide-free bainite. Therefore, the presented welding methods can be useful for planning the welding process of bainitic steels with a similar chemical composition”.

2.  "Table 2 includes some results of friction stir welding. While this content is explained briefly in section 3 of "Review of Welding Methods," in Section 4 of "Analysis of microstructure changes after the welding process," no discussion about Stir-Zone and thermo-mechanical affected zone is provided."

Response 2: Friction Stir Welding is described only in section 3 due to the microstructure analysis included in the literature references. The authors described the microstructure using only scanning microscopy, which is insufficient for such structures. Among other, they state different content of austenite in various zones without performed EBSD or XRD investigations. Therefore, a detailed analysis in section 4 is not possible. The authors of these articles mostly focused on the Friction Stir Welding process technology.

If the answers are not exhaustive, we will gladly answer the next questions.

Thank you for your consideration.

Sincerely

Aleksandra Królicka

Reviewer 2 Report

Remarks to the authors:

Although bainitic steels were extensively studied in the course of the past century, little is known about their behavior in today’s manufacturing processes. Here, Królicka and colleagues investigate the weldability of nano-strucured carbide-free bainite. The authors review, quite comprehensively, the influence of different welding technics on microstructure as presented by different researchers in the field. They also covered a number of solutions that help maintain the high mechanical properties desired when exploiting bainites. These novel insights are likely to be of significant interest to the welding community. This paper, in my opinion, can be published but after addressing the following minor suggestions:

·        Paragraph 2:

1.     In general, the paragraph was brief. The influence of the welding metal was swiftly dismissed after only three citations.

2.    Lines 64 and 65 mention: “The presence of unstable blocky austenite reduces the plastic properties of welds (impact toughness) due to the induction of martensitic transformation under loads”

This statement need more elaboration since it is well documented that a stress induced austenite-martensite transformation increase plasticity (TRIP effect). Furthermore, Table 1 mentions 14% elongation for that same case.

·        Paragraph 3:

In rotary impacting trailed welding section: it would be helpful to mention why deforming the austenite accelerates bainite formation (why some authors suggested this technic in the first place)

·        Paragraph 4:

Annotations in Figures presenting micrographs can be helpful: i.e. indicating blocky and film austenite in figure 4-a and pointing out the bainite mentioned in line 247 in figure 4-d (“blocky austenite that decomposes into a fine nanostructured bainite with a different crystallographic orientation than the nearby bainite”)

·        The text needs to be proofread for typos, e.g.:

1.    2 comas in Line 45: “welding process, , as a result”

2.    Line 111: “method uses a firs impacting head placed”

3.    In table 2, the colon presenting process parameters for reference [22]: “NapiÄ™cie: 18 V”

4.  The figures “Scheme of welded joint” and the “Thermal cycles simulation” are both numbered “Figure 3”

5.    Line 282: “ In the Fuzion zone”

6.    Paragraph numbering in the conclusion: 5.1 and 5.2 instead of 4.1 and 4.2.

Author Response

Dear Reviewer,

Thank you for your insightful review of our manuscript and few important comments. We hope that with their help we were able to improve the quality of the article. We have compiled your comments and our responses below:

1. Paragraph 2: In general, the paragraph was brief. The influence of the welding metal was swiftly dismissed after only three citations.

Response 1: A short chapter on design of weld-metals is associated with a small number of scientific reports in this area. We found only three research articles about nanobainitic welding materials. This subject requires further research, which has been highlighted in the perspectives.

2. Paragraph 2: Lines 64 and 65 mention: “The presence of unstable blocky austenite reduces the plastic properties of welds (impact toughness) due to the induction of martensitic transformation under loads”

This statement need more elaboration since it is well documented that a stress induced austenite-martensite transformation increase plasticity (TRIP effect). Furthermore, Table 1 mentions 14% elongation for that same case.

Response 2: The morphology of austenite significantly affects its stability. Unstable blocky austenite increases elongation due to the TRIP effect (Dong, B.; Hou, T.; Zhou, W.; Zhang, G.; Wu, K. The Role of Retained Austenite and Its Carbon Concentration on Elongation of Low Temperature Bainitic Steels at Different Austenitising Temperature. Metals 2018, 8, 931-942). In contrast, stable austenite with film-like morphology increases strength and impact toughness. Despite the higher elongation during dynamic loads, steel with more content of blocky austenite will show significantly lower impact toughness nanostructured bainite. (Tsai, Y.T.; Chang, H.T.;, Huang, B.M.; Huang, C.Y.; Yang, J.R. Microstructural characterization of Charpy-impact-tested, Mater. Charact. 2015, 107, 63-69).  For this reason, it is more preferable to increase the volume of stable film-like austenite than unstable blocky austenite.

Line 64-65 has been specified: “The presence of unstable blocky austenite reduces the impact toughness of welds due to the induction of martensitic transformation under loads [15]”.

The following sentence has been added to section 4: Line 296: “The presence of less stable, coarse blocky austenite in both the FZ and HHAZ zones is the reason for a significant reduction of impact toughness in relation to the base material [15,42-43]. On the other hand, blocky austenite increases elongation due to the TRIP effect [6]. A more preferred morphology is stable film-like austenite because it increases strength and fracture toughness. For this reason, a large amount of blocky retained austenite in FZ and HHAZ is undesirable.

3. Paragraph 3: In rotary impacting trailed welding section: it would be helpful to mention why deforming the austenite accelerates bainite formation (why some authors suggested this technic in the first place) 

Response 3: It has been added in the text, with additional references: “The authors proposed such a process due to the fact that fine austenite grains will accelerate the nanobainitic transformation [23-24] in the case of the material used in this research[22]. Acceleration of bainitic transformation will shorten the regeneration time, which will reduce the cost of heat treatment and increase the process's technological efficiency.

4. Paragraph 4: Annotations in Figures presenting micrographs can be helpful: i.e. indicating blocky and film austenite in figure 4-a and pointing out the bainite mentioned in line 247 in figure 4-d (“blocky austenite that decomposes into a fine nanostructured bainite with a different crystallographic orientation than the nearby bainite”)

Response 4: Thank you for your very valuable comment. This would certainly improve the readability of the figures. Unfortunately, the figures were included in the original version, with the publisher's permission.

5. The text needs to be proofread for typos, e.g.:

2 comas in Line 45: “welding process, , as a result” Line 111: “method uses a firs impacting head placed” In table 2, the colon presenting process parameters for reference [22]: “NapiÄ™cie: 18 V” The figures “Scheme of welded joint” and the “Thermal cycles simulation” are both numbered “Figure 3” Line 282: “ In the Fuzion zone” Paragraph numbering in the conclusion: 5.1 and 5.2 instead of 4.1 and 4.2                   

Response 5Thank you for your proofread help. All comments have been corrected in the text.

If the answers are not exhaustive, we will gladly answer the next questions.

Thank you for your consideration

Sincerely

Aleksandra Królicka

Reviewer 3 Report

line 54: there two commas are use

line 94: ausenite correct austenite

Fig.7: I am not sure if the figure shows dendritic structure

Author Response

Dear Reviewer,

Thank you for your review of our manuscript and your proofread. We have compiled your comments and our responses below:

1. line 54: there two commas are use

Response 1: corrected in the text

2. line 94: ausenite correct austenite

Response 2: corrected in the text

3. Fig.7: I am not sure if the figure shows dendritic structure

Response 3: The numbering of the figures has been corrected. The dendritic structure (according to the authors [19]) is now shown in Fig. 8.

If the answers are not exhaustive, we will gladly answer the next questions.

Thank you for your consideration.

Sincerely

Aleksandra Królicka

Round 2

Reviewer 1 Report

The content of the revised manuscript is almost identical to the original version.